# Improved Mucoadhesion, Permeation and In Vitro Anticancer Potential of Synthesized Thiolated Acacia and Karaya Gum Combination: A Systematic Study

**DOI:** 10.3390/molecules27206829

**Published:** 2022-10-12

**Authors:** Neha Munot, Ujjwala Kandekar, Chaitali Rikame, Abhinandan Patil, Poulomi Sengupta, Shabana Urooj, Anusha Bilal

**Affiliations:** 1Technical Lead, HCL Technologies, Chennai 600058, Tamil Nadu, India; 2Department of Pharmaceutics, Rajarshi Shahu College of Pharmacy and Research, Tathwade, Pune 411033, Maharashtra, India; 3Department of Pharmaceutics, Sinhgad Technical Education Society’s Smt. Kashibai Navale College of Pharmacy, Pune 411048, Maharashtra, India; 4D. Y. Patil College of Pharmacy, D. Y. Patil Education Society, Deemed to Be University, Kolhapur 416006, Maharashtra, India; 5Physical/Material Chemistry Division, National Chemical Laboratoty (CSIR-NCL), Pashan, Pune 411008, Maharashtra, India; 6Department of Chemistry, Indrashil University, Kadi 382740, Gujarat, India; 7Department of Electrical Engineering, College of Engineering, Princess Nourah Bint Abdulrahman University, P.O. Box 84428, Riyadh 11671, Saudi Arabia; 8Department of Food Processing & Technology, School of Vocational Studies and Applied Sciences, Gautam Buddha University, Greater Noida 201312, Uttar Pradesh, India

**Keywords:** mucoadhesion, thiolated gums, ivabradine HCl, buccal tablets, enhanced permeation

## Abstract

Thiolation of polymers is one of the most appropriate approaches to impart higher mechanical strength and mucoadhesion. Thiol modification of gum karaya and gum acacia was carried out by esterification with 80% thioglycolic acid. FTIR, DSC and XRD confirmed the completion of thiolation reaction. Anticancer potential of developed thiomer was studied on cervical cancer cell lines (HeLa) and more than 60% of human cervical cell lines (HeLa) were inhibited at concentration of 5 µg/100 µL. Immobilized thiol groups were found to be 0.8511 mmol/g as determined by Ellman’s method. Cytotoxicity studies on L929 fibroblast cell lines indicated thiomers were biocompatible. Bilayered tablets were prepared using Ivabradine hydrochloride as the model drug and synthesized thiolated gums as mucoadhesive polymer. Tablets prepared using thiolated polymers in combination showed more swelling, mucoadhesion and residence time as compared to unmodified gums. Thiol modification controlled the release of the drug for 24 h and enhanced permeation of the drug up to 3 fold through porcine buccal mucosa as compared to tablets with unmodified gums. Thiolated polymer showed increased mucoadhesion and permeation, anticancer potential, controlled release and thus can be utilized as a novel excipient in formulation development.

## 1. Introduction

Gums are usually utilized as mucoadhesive polymers, as they are regarded as safe by the FDA and are biodegradable, biocompatible and their processing is environmental friendly and economical [1]. A variety of gums are widely used alone or in combination with other gums as an emulsifying agent, suspending agent, binder in tablets, demulcent and emollient in cosmetics, dental adhesive, bulk laxative and mucoadhesive, etc. [2,3,4]. Gum acacia is the dried gummy exudation obtained from the stem and branches of *Acacia arabica* or closely related species of *Acacia* (*Acacia senegal* and *A. seyal*) belonging to family *Leguminosae* [5]. Gum karaya is the dried exudate obtained from *Sterculia urens* Roxd. and other related species of *Sterculia* belonging to family *Malvaceae* [6]. Both these gums had been explored in the food and pharmaceutical industries for a variety of applications [7,8]. Various modifications have been carried out on gums in order to improve their efficacy and functionality. Hassanzadeh–Afruzi et al. synthesized acacia gum-*grafted*-polyamidoxime—copper ferrite magnetic nanoparticles by graft copolymerization reaction. This ecofriendly catalyst was used for synthesis of pyrazolopyridine derivatives [9] and it was also used for repairing the damage of chlorpyrifos pesticide from contaminated water due to its superabsorbent and magnetic nature [10]. E.N. Zare et al. prepared biocomposite films using gum tragacanth by solvent casting technique. These films had antibacterial and antioxidant activity and were used as food packaging material [11].

One of the important applications of these gums in the pharmaceutical industry is their usage as mucoadhesive agents [12]. First generation mucoadhesive polymers form weak noncovalent bonds such as hydrogen bonds, Vander Waal’s interaction and ionic interaction with the mucus glycoproteins; therefore, they cannot localize a drug delivery system at a desired site for a longer period of time and can be carried away due to gastrointestinal transit. This will decrease bioavailability and increase dosing frequency of the drug, which has a narrow absorption window, shorter half-life and ultimately decreases patient compliance. Hence, there is a need for new generation mucoadhesive polymers which will reside at a target site for longer period of time and control the release of the drug so that only small amount of drug will get released at a time and will be absorbed at a faster rate due to the residence of a dosage form at its absorption site.

Thiols modified of polymers (thiomers) are capable of forming covalent bonds, leading to improved mucoadhesion. Thiomers are capable of forming intra- and interchain disulfide bonds within the polymeric network, leading to strongly improved cohesive properties and stability of drug delivery systems [13]. Due to the formation of strong covalent bonds with mucus glycoproteins, thiomers show the strongest mucoadhesive properties of all currently tested polymeric excipients via thiol disulfide exchange reaction and an oxidation process [14]. Besides, thiolated polymers have the ability to control the drug delivery, enhancement of drug permeation [15], and have an in situ gelling property, etc. [16].

In the present research work, gum acacia and gum karaya were selected for thiol modification as they are of natural origin and anionic polymers [17]. Thiolation of gum acacia and gum karaya in combination was carried out by using thioglycolic acid. This modification resulted in improved mucoadhesion, swelling, enhanced permeation and controlled the release of drugs as compared to a mixture of unmodified gums (PM). Anticancer potential of thiolated gums was established on cancerous HeLa cell lines. Ivabradine HCl was chosen as a model drug because it has low oral bioavailability (near about 40%). It has a shorter half-life of 2–3 h and the dosing frequency is twice a day. Development of the buccal drug delivery of Ivabradine Hydrochloride is one of the alternative routes of administration to avoid first pass metabolism. Formulation of buccal tablets of Ivabradine HCl using synthesized thiolated gum (TRM) increased bioavailability of Ivabradine HCl due to improved mucoadhesion and permeation enhancement.

## 2. Result and Discussion

### 2.1. Synthesis of Thiolated Polymer Using Combination of Gum Acacia and Gum Karaya: Thiolated Reaction Mixture (TRM)

Initially, acacia was thiolated individually and characterized. Thiolated acacia swelled rapidly within 2 h and it eroded after that. Also, it possessed poor mucoadhesion strength of about 32.54 gm, and residence time was found to be 36 ± 2 min. Hence, it was concluded that individual thiolated acacia cannot be used for formulation development. Hence, thiol modification of gum acacia and gum karaya in combination was thought of for the first time, which could improve efficacy of developed formulation.

Thiolation of gum acacia and gum karaya in combination was achieved by formation of ester bond between galactouronic acid moieties (hydroxyl groups) in PM and carboxyl groups in TGA (Figure 1). The average yield of this synthesis was found to be 56%. The product was washed with methanol to remove unreacted TGA as TGA has solubility in methanol. Then the product was recrystallized to produce pure TRM. It was found to be white to buff in color. The developed product possessed good flow properties and Carr’s index was found to be 12%, hence it could be used as an excipient for directly compressible tablets.

### 2.2. Characterization of Synthesized Thiomer (TRM)

#### 2.2.1. Determination of Thiol Group Content (Ellman’s Method)

Thiol group content in modified polymers was determined by Ellman’s method. TRM was found to contain 0.8511 mmol of thiol groups/g, whereas that of thiolated acacia was 0.2408 mmol/g. This indicated better modification in TRM as compared to individual thiolated acacia (TA) and hence TRM exhibits better potential to react with mucous glycoproteins resulted in enhanced mucoadhesion, permeation and controlled drug release.

#### 2.2.2. Fourier Transform Infra-Red Spectroscopy (FT-IR)

As seen in Figure 2, PM shows broad peak of –OH group at 4002 cm^−1^, C-H stretch at 2921 cm^−1^, COO asymmetric stretch at 1730 cm^−1^ and COO symmetric stretch at 1375 cm^−1^ [18]. FT-IR spectrum of TRM showed all the peaks found in PM, but an additional peak at 2360 cm^−1^ due to –SH stretch and more intensity COO stretch at 1748 cm^−1^ which confirms modification of PM and formation of TRM [13].

#### 2.2.3. Differential Scanning Calorimetry (DSC)

As seen in Figure 3, DSC thermogram of PM show two endothermic peaks at 66.17 and 199.19 °C representative of mixture of gum acacia and karaya, respectively. Modification of PM can be confirmed as these peaks were found to be absent in DSC thermogram of TRM which showed a single, sharp endotherm at 69.48 °C corresponding to its melting point.

#### 2.2.4. X-ray Diffraction-XRD Studies

Figure 4 shows X-ray Diffraction spectra of PM and TRM. No intense peaks were observed in a pure gum sample indicative of amorphous nature of gums. These results were found to be in agreement with studies carried out for gums [19,20,21], while X-ray diffractogram of TRM show characteristic intense peaks indicating greater degree of crystallinity as compared to pure gums [22].

#### 2.2.5. Cytotoxicity Study

As TRM is a new polymer, it was important to determine its safety and biocompatibility. Healthy L929 fibroblast cell lines were used for this study as these cell lines are recommended by USP to determine biological reactivity of newly synthesized molecules [23]. Cytotoxicity studies were carried out on L929 fibroblast cell lines to assess cell viability and thus safety and biocompatibility of thiolated polymer. This colorimetric assay measures the reduction of yellow 3-(4,5-dimethylthiazole-2-yl)-2,5diphenyl tetrazolium bromide (MTT) by mitochondrial succinate dehydrogenase to purple coloured formazan [15]. As seen in Figure 5a, more than 75% cells were viable at all the concentration ranges of PM and TRM which were studied (selected concentration ranges were comparable to concentrations used for cytotoxicity studies for other well established thiomers). Thus, it can be concluded that thiol modification does not form any cytotoxic compound and can be regarded as safe and biocompatible.

#### 2.2.6. Anticancer Activity

As seen in Figure 5b, TRM showed anticancer activity in vitro, as more than 60% of human cervical cell lines (HeLa) were inhibited at a concentration of 5 µg/100 µL. Thiol modification can also lead to anticancer activity is reported for the first time. The mechanism for this activity is unclear but it can be predicted to be as follows:

The basis of enhanced mucoadhesion in thiomers is the interaction of thiol group on polymer backbone with sulphydryl groups on cysteine residues in mucous glycoprotein. This indicates that they have the ability to form covalent bonds with cysteine. DNA topoisomerase IIα (Top2α), a tumor marker is highly up-regulated in transformed and cancerous cells. Covalent modification of several cysteine residues on human Top2α leads to Top2α cleavage complexes leading to apoptosis. Hence, it can be said that thiol groups on thiomers can covalently bind to cysteine residues on Top2α, leading to its cleavage and inhibition and ultimately resulting in the apoptosis of cells and thus have anticancer activity. These results are in accordance with activity of Top2α targeting drugs, such as doxorubicin and mitoxantrone, which are known to trap lethal Top2α-DNA covalent adducts, termed Top2α cleavage complexes, which are responsible for the antitumor activity of Top2-targeting drugs [24].

Thus, thiolated polymers can be used in future for targeting anticancer drugs to the cancerous cells and may show synergistic activity.

### 2.3. Formulation of Bilayered Buccal Tablets of Ivabradine HCl

Bilayered buccal tablets of Ivabradine hydrochloride were formulated using PM and TRM separately. Ethyl cellulose was used as backing layer to have unidirectional release of drug through buccal mucosa, so as to avoid first pass metabolism and thus increased bioavailability. Drug was dispersed uniformly in TRM and these bilayered tablets were evaluated to determine the effect of thiol modification of gums on different properties such as swelling, mucoadhesive strength, residence time on buccal mucosa, controlled release of the drug and permeation enhancement of the drug leading to increased bioavailability of drug (Table 1).

#### 2.3.1. Swelling Study

As thiomers form disulfide bonds within the polymer, their hydration capacity increases. Swelling behavior has a great impact on adhesive and cohesive properties of polymers and can lead to controlled release of drug as the diffusion path length increases. The results of swelling index are shown in Table 2. Batches of F1–F3 containing PM showed rapid swelling in pH 6.8 phosphate buffer followed by its erosion within 6 h. Tablets prepared using TRM, i.e., batches F4–F6 swelled gradually for 24 h, and their integrity was maintained. This can be attributed to the covalent attachment of TGA to polymer which results in inter and intramolecular disulfide bonds within the thiomer due to which tablets of batch F4 to F6 could absorb water in multiples of their own weight. Swelling index increased as the amount of thiolated polymer increased, but as seen in batch F6, where excess amount of TRM was added, it resulted in more cross linking of polymer and thus less swelling as compared to batch F5 [22].

#### 2.3.2. Mucoadhesive Strength

As seen in Table 2, mucoadhesive strength of tablets prepared using TRM, i.e., batches F4 to F6 was found to be more than batches F1 to F3, which were prepared using unmodified gum, PM. An increase in mucoadhesive strength was almost 2 fold (in batch F5) due to formation of disulfide bonds (covalent bonds) with the mucus glycoproteins in porcine buccal mucosa instead of forming weak noncovalent bond as in case of PM.

#### 2.3.3. Ex Vivo Residence Time

Adhesion time is a critical factor to design effective mucoadhesive formulation. The mucoadhesive property of gum acacia and gum karaya is due to number of hydroxyl groups in its structure. These –OH groups form weak non-covalent bonds such as hydrogen bonds, Van der Waal’s interaction or ionic interaction with the mucus glycoproteins hence show weak mucoadhesion. In contrast, TRM has sulfhydryl (-SH) groups in its structure which form strong covalent disulfide bonds with the mucus glycoproteins hence show strong mucoadhesion in comparison to the PM. Residence time of the tablet of F2 in phosphate buffer pH 6.8 was found to be 2 h while that of batch of F5 was found to be more than 10 h.

#### 2.3.4. In Vitro Drug Release Study

The drug release profile of tablets with PM (batch F2) and TRM (batch F5) are shown in Figure 6a. Release of Ivabradine HCl from tablets of batch F2 was 100% within 4 h, whereas tablets of batch F5, released the drug slowly and gradually up to 24 h. Controlled drug release from TRM matrix tablets (batch F5) was found to be another advantage for a modified polymer to be used as an excipient in novel drug delivery systems. One reason for the comparatively slow release of drug from tablet containing thiolated polymer may be the formation of inter/intrachain disulfide bonds which holds the polymer chain together, increases crosslinking of the polymer and hence enhance cohesiveness of the matrix tablet. Also, in swelling studies it was proved that thiolated polymer matrix tablet had greater swelling index and showed no disintegration and erosion throughout the study. This probably increases diffusion path length and controls rate of diffusion of the drug molecules through the polymer matrix.

#### 2.3.5. Ex Vivo Permeation of Ivabradine Hydrochloride Form Buccal Tablets

Thiol modification leads to enhanced permeation of drugs, as revealed from in vitro drug permeation studies carried out on porcine buccal mucosa using Franz diffusion cells. Figure 6b indicates that at the end of 24 h, drug permeated from tablets prepared using PM and TRM was found to be 28.29% and 64.78%, respectively. These results were compared with plain drug solution (11 mg) in water. As Ivabradine HCl belongs to BCS class I, it has high solubility and permeability. Hence, after 24 h, 58.40% of the drug was permeated. Drug permeation of tablets belonging to batch F2 was low as the drug was embedded in the matrix of gums which retarded its release as well as permeation. In contrast, tablets prepared using TRM showed enhanced permeation. The mechanism for less permeation of drugs through mucosa is due to formation of tight junctions because of dephosphorylation of tyrosin residues by enzyme protein tyrosine phosphatase (PTP). Thiol modification inhibits PTP, thereby preventing dephosphorylation of tyrosine residues, ultimately leading to opening of tight junctions and increased permeation of drugs through mucosa [25]. Also, thiomers have high molecular mass as compared to other low molecular mass permeation enhances, and hence they will not be absorbed from the mucosal tissue and therefore can act for a longer time period while systemic side effects can be excluded.

## 3. Materials and Methods

### 3.1. Synthesis of Thiolated Polymer Using Combination of Gum Acacia and Gum Karaya: Thiolated Reaction Mixture (TRM)

Gum acacia (2.5 g) and gum karaya (2.5 g) were dispersed in 70 mL of cold water. The dispersion was esterified by addition of 2.5 mL of thioglycolic acid (80%) and 2 mL of 7N hydrocholic acid. Further, mixture was allowed to react at 70 °C for 4 h. The resultant mixture was poured in 500 mL methanol. White precipitate of thiolated polymer so obtained was washed twice with methanol, recrystallized with ethanol and dried under IR lamp [13].

### 3.2. Characterization of Synthesized Thiomer (TRM)

TRM was characterized for flow properties and other various parameters as mentioned below and results were compared with plain mixture of gums (PM-mixture of unmodified acacia and karaya gum in the ratio 1:1).

#### 3.2.1. Determination of Thiol Group Content (Ellman’s Method)

Degree of modification, i.e., the amount of free thiol groups immobilized on TRM was determined by Ellman’s reagent [26]. Thiolated polymer (50 mg) was dispersed in 25 mL of distilled water. An aliquot of 2.5 mL of polymer dispersion was diluted with 2.5 mL of 0.5 M phosphate buffer (pH 8.0) and allowed to react with 5 mL of Ellman’s reagent, i.e., 5,5′-Dithio-bis-(2-nitrobenzoic acid) (DTNB, 0.03% *w/v* in 0.5 M phosphate buffer, pH 8.0) for 2 h at room temperature in the dark. The absorbance of reaction mixture was noted at 412 nm by using UV Spectrophotometer (Jasco V-630). The number of thiol group was calculated from standard curve prepared by reacting an increasing amount of L-cysteine with Ellman’s reagent.

#### 3.2.2. Fourier Transform Infra-Red Spectroscopy (FT-IR)

The presence of the thiol group in TRM was studied via FT-IR spectroscopy. Pure sample of Plain gum mixture (PM) and thiolated reaction mixture (TRM)were scanned by Fourier Transform Infra-Red spectrophotometer (Shimadzu Affinity-1). KBr pellets of samples were prepared by applying pressure of 10 ton and FTIR spectra was recorded in a range of 4500–500 cm^−1^ [27,28].

#### 3.2.3. Differential Scanning Calorimetry (DSC)

The thermal behavior of pure gum sample (PM) and modified gum (TRM) was studied by using a differential scanning calorimetry. Sample (1.01 mg) was crimped in a standard pan and heated in a temperature range of 30–450 °C at a heating rate 15 K per minute in a nitrogen atmosphere in differential scanning calorimeter (Perkin Elemer) [29].

#### 3.2.4. X-ray Diffraction (XRD)

X-ray diffraction was carried out to investigate the nature of the PM and TRM using X-ray diffractometer (Shimadzu-binary). The polymer was scanned from 10° to 80° diffraction angle (2Q) under the following measurement conditions. Source—nickel filtered Cu-K alpha radiation, Voltage—40 kV, Current—30 mA, Scan speed—0.02 min^−1^ [30].

#### 3.2.5. Cytotoxicity Studies

Cytotoxicity studies were carried out on L929 fibroblast cell lines to assess cell viability, safety and biocompatibility of thiolated polymer. The studies were carried out as per the method reported by Sallah et al. (2020) with slight modification. PM and TRM were sterilized for 30 min by UV exposure inside biosafety cabinet. Samples were initially dissolved in sterile dimethyl-sulphoxide (DMSO) and further dilutions (10, 5, 2.5, 1.25 μg/100 μL) were carried out using complete Dulbecco’s Modified Eagle Medium (DMEM). Healthy L929 fibroblast cell lines (passage number 59) were maintained using complete DMEM. These cells (10,000) were plated in each well of a 96 well plate. It was incubated in a 5% CO_2_ environment at 37 °C for 1 day. Samples of above concentrations were added in triplicates. After 24 h of incubation, 100 μL of 200 μM resazurin solution (in complete DMEM media) was added. It was then further incubated for 6 h in 5% CO_2_ at 37 °C. From each well 100 μL media was taken out and read in a plate reader (excitation 530 to 560 nm and emission at 590 nm) [31]. The percentage of cell viability was calculated via following formula:% Cell viability = Average absorbance of test/Average absorbance of control × 100(1)

#### 3.2.6. Anticancer Activity

Anticancer activity of synthesized thiomer was studied for cancerous HeLa cell lines. PM and TRM were sterilized by 30 min UV exposure inside the biosafety cabinet. Samples were initially dissolved in sterile DMSO and further dilutions (10, 5, 2.5, 1.25 μg/100 μL) were carried out using complete DMEM. HeLa cervical cancer cell lines (passage number 89/90) were maintained using complete DMEM. Cells (10,000) were plated in each well of a 96 well plate. It was incubated in 5% CO_2_ at 37 °C for 1 day. Samples of above concentrations were added in triplicates. After 24 h incubation in 5%, CO_2_ at 37 °C, 100 μL of 200 μM resazurin solution (in complete DMEM media) was added. It was incubated for 6 h in 5% CO_2_ at 37 °C. The entire plate was read in a plate reader (excitation 530 to 560 nm and emission at 590 nm). The percentage of cell viability was calculated using Formula (1).

### 3.3. Formulation of Bilayered Buccal Tablets of Ivabradine HCl

Plain mixture and thiolated reaction mixture of gum acacia and gum karaya were directly compressed separately with 11 mg of drug to form flat faced tablets using 8 mm punch of Rotary compression machine (Lab Press) at a constant compression force. These tablets were used as a core layer. The core layer and backing layer of ethyl cellulose were sequentially compressed, as shown in Table 2 by indigenously developed and standardized dies and punches on tablet compression machine.

### 3.4. Evaluation of Bilayer Buccal Tablets of Ivabradine Hydrochloride

#### 3.4.1. Swelling Study

Tablets of formulation batches F1 to F6 were weighed initially (W_0_) and were placed in a 100 mL of phosphate buffer pH 6.8 at 37 °C. At a regular time interval of 1 h till 10 h and then at 24th h excess surface water was removed by using blotting paper very carefully and these swollen tablets were again weighed (W_t_) [22]. The Swelling index is calculated by using following formula:Swelling index = W_t_ − W_0_ / W_t_ × 100(2)

#### 3.4.2. Mucoadhesive Strength

To perform the ex vivo mucoadhesive strength fresh porcine buccal mucosa was obtained from a local slaughterhouse and used within 2 h of slaughter. The mucosal membranes were separated by removing the underlying fat and loose tissues. The membranes were washed with distilled water and transferred to buffer pH 6.8 phosphate buffer. The membranes were then fastened to Perspex block with cyanoacrylate glue and the block was placed in a holder fixed firmly to the base of the mucoadhesive assembly. The tablets prepared using PM and TRM were then attached to a rubber cork separately. This rubber cork was suspended above the Perspex block with the mucoadhesive layer facing the membrane with the help of double side tape. The rubber cork was tied to a cylindrical vessel at three corners using a string of good mechanical strength. The string was coiled around a pulley and the cylindrical vessel was suspended freely. A burette filled with water was placed above the cylindrical vessel. The block attached with membrane was filled with 100 mL of pH 6.8 phosphate buffer and the cork was lowered until the tablet came into contact with the buccal mucosal membrane. The contact time of the tablet with the mucosal membrane was 10–20 s. along with the application of optimum external pressure. Water from the burette was added drop wise in the suspended vessel till the tablet got dislodged from the mucosal membrane. The volume of water (mL) required to dislodge the tablet from the mucosal membrane was correlated with the mucoadhesive strength (gm) and the calculation was done by using following formula [32].
Force of adhesion (N) = Mucoadhesive strength ÷ 1000 × 9.81(3)

#### 3.4.3. Ex Vivo Residence Time

This test was carried out as per the method described by Szekalska et al. (2019) with slight modification. Tablets of different batches (F2 and F5) were attached to freshly excised porcine buccal mucosa separately, which was fixed on a basket of USP type Ι apparatus (Labindia DS 8000). Thereafter the basket was allowed to dip in the respective dissolution media (500 mL of phosphate buffer of pH 6.8) at 37 ± 0.5 °C, 100 rpm. The time required for detachment of tablet was recorded as ex vivo residence time [33].

#### 3.4.4. In Vitro Drug Release Study

In vitro release of ivabradine hydrochloride from tablets of formulation batches F2 and F5 was studied. The study was carried out using USP dissolution testing apparatus Type II (Labindia DS 8000) containing 100 mL of phosphate buffer pH 6.8 as a dissolution media, rotating at the speed of 100 rpm. At a definite interval, 5 mL aliquot was removed and replaced by the same medium to maintain sink conditions. This 5 mL of aliquot was diluted to 10 mL with respective media, and absorbance was taken at 287 nm using UV Spectrophotometer [27].

#### 3.4.5. Ex Vivo Permeation of Ivabradine Hydrochloride form Buccal Tablets

The porcine buccal mucosa was mounted between the donor and receptor compartment of the standard Franz diffusion cell with a diffusion area of 30.02 cm^2^. Phosphate buffer pH 6.8 (10 mL maintained at 37 °C) was added in the acceptor compartment which was continuously stirred at 600 rpm using a magnetic stirrer. The tablets of batches F2 and F5 were wetted with 1 mL of phosphate buffer and were placed into the donor compartment. The amount of drug permeated through the membrane was determined by removing aliquots (3 mL) from the receptor compartment and by replacing the same volume of buffer at every hour for 8 h and then 23rd and 24th h. The amount of drug permeated was determined using a UV Spectrophotometer at 287 nm. The results were compared with permeation of solution of Ivabradine hydrochloride through buccal mucosa [34,35,36].

## 4. Conclusions

Thiol modification of the combination of gum acacia and karaya was successfully carried out for the first time, and this polymer can be utilized for mucoadhesive drug delivery with permeation enhancement of various drugs with poor bioavailability due to hepatic metabolism. Here, Ivabradine HCl was used as a model drug. Tablets prepared using thiol modified gums showed better mucoadhesion, swelling, improved ex vivo residence time, controlled the release of drug for 24 h and enhanced permeation of drug through buccal mucosa as compared to tablets prepared using unmodified gums. Thiol modification led to the selective killing of cancerous cells (HeLa cells), whereas normal cells (Human fibroblast L929) were spared, thus indicating their safety and biocompatibility for normal cells and tumour cell targeting potential. Anticancer activity of thiolated polymers was explored for the first time. These results indicate that this novel polymer can be used as an excipient for development of drug delivery with improved performance.

## Figures and Tables

**Figure 1 molecules-27-06829-f001:**
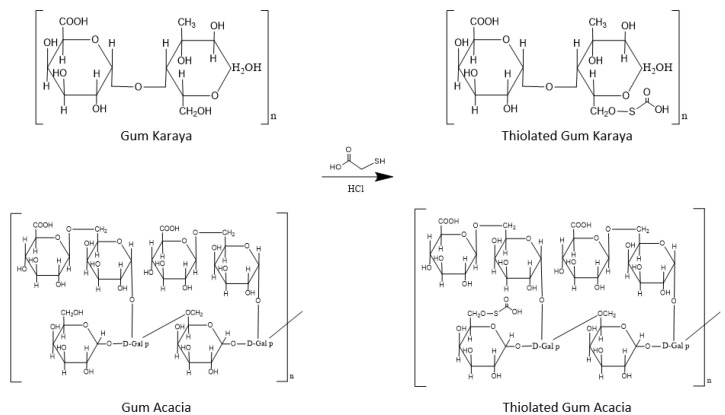
The synthesis of thiolated gum Karaya and Acacia.

**Figure 2 molecules-27-06829-f002:**
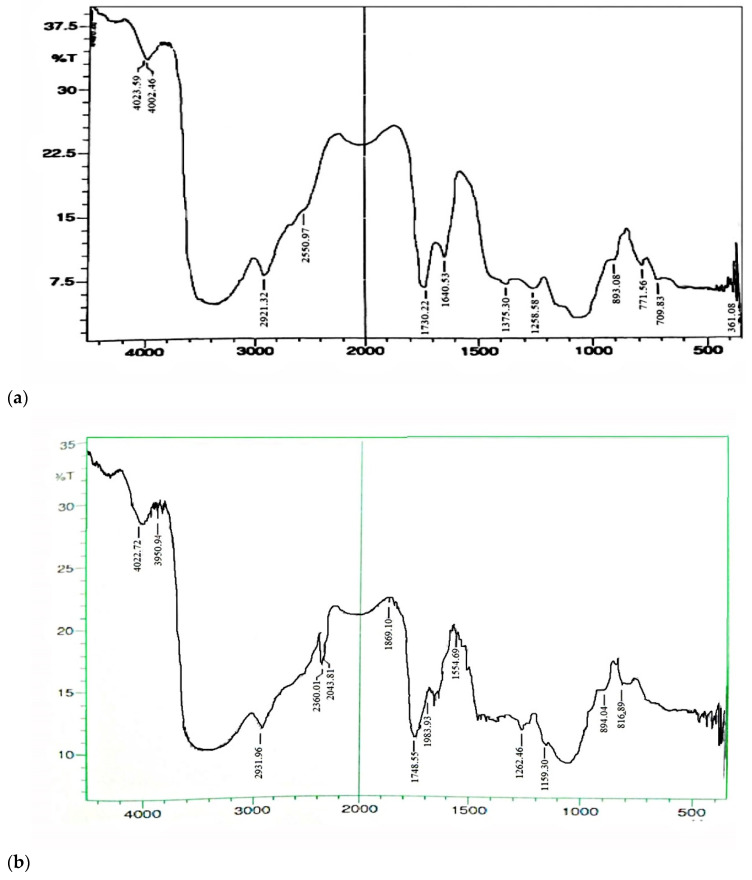
FT-IR spectrum of (**a**) Plain mixture and (**b**)TRM of gum acacia and karaya.

**Figure 3 molecules-27-06829-f003:**
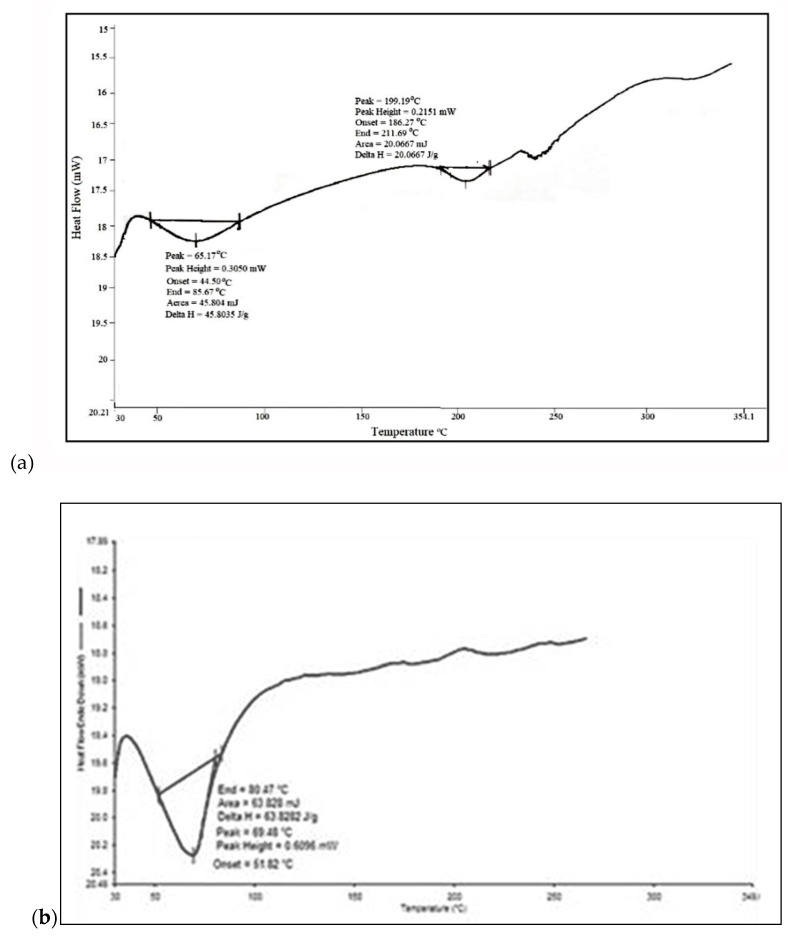
ADSC thermogram of the Plain mixture (**a**) and Thiolated reaction mixture (**b**).

**Figure 4 molecules-27-06829-f004:**
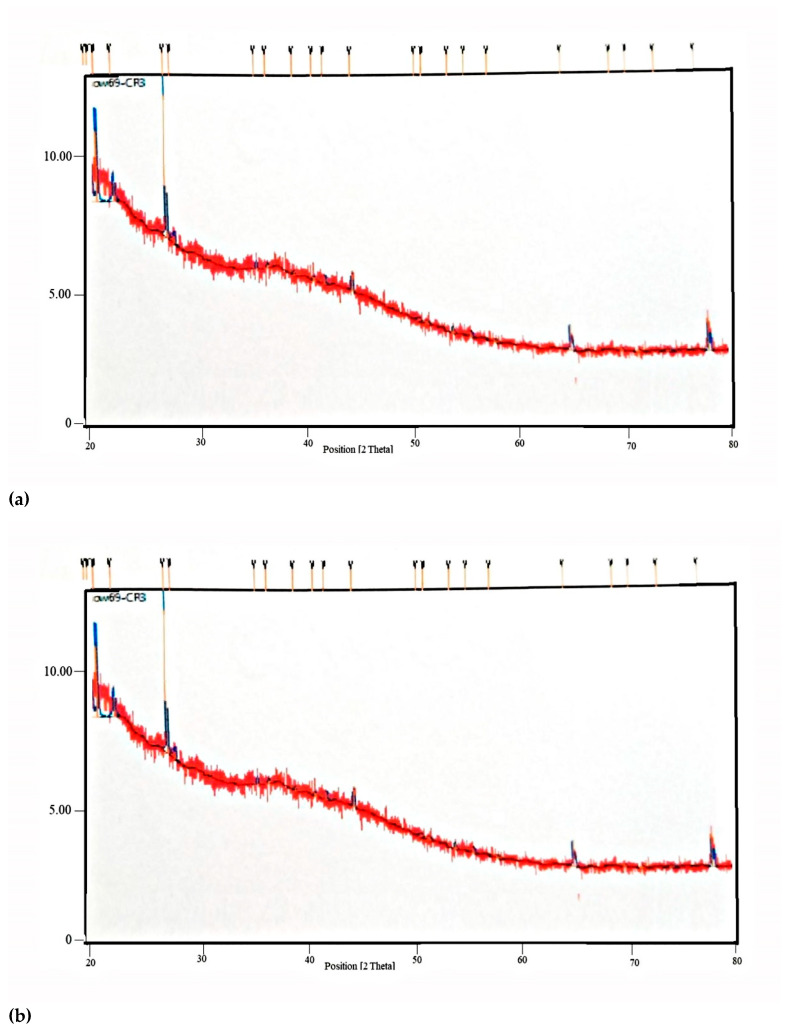
X-ray diffraction (XRD) of (**a**) Plain mixture and (**b**) TRM of gum acacia and gum karaya.

**Figure 5 molecules-27-06829-f005:**
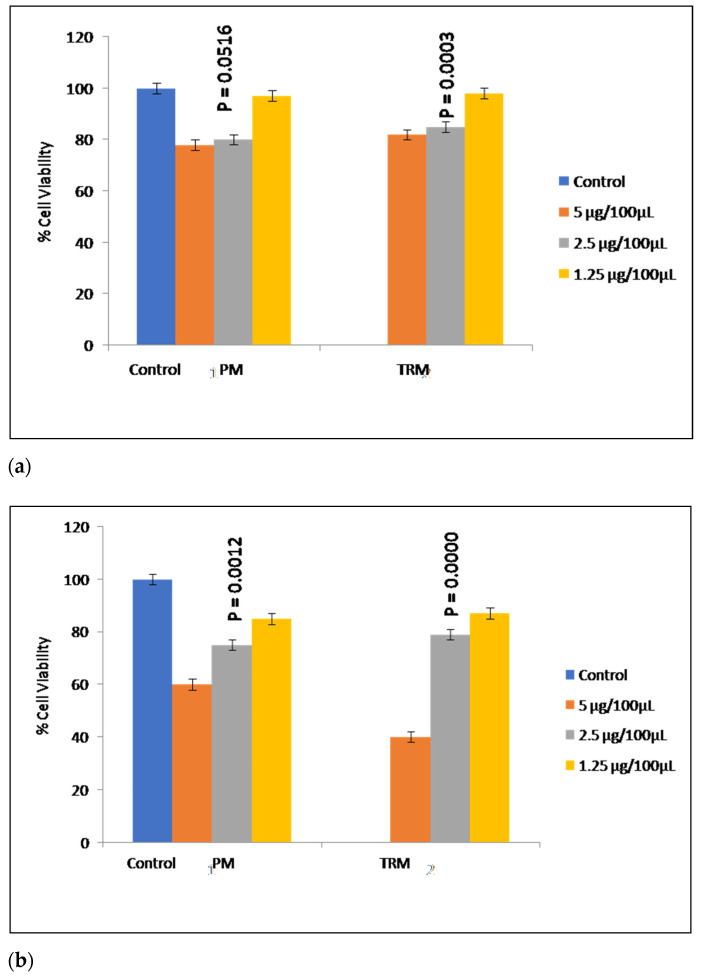
(**a**) A ctotoxicity study of PM and synthesized TRM on L929 fibroblast cell lines; (**b**) in vitro anticancer studies of PM and synthesized TRM on human cervical cancer HeLa cell lines.

**Figure 6 molecules-27-06829-f006:**
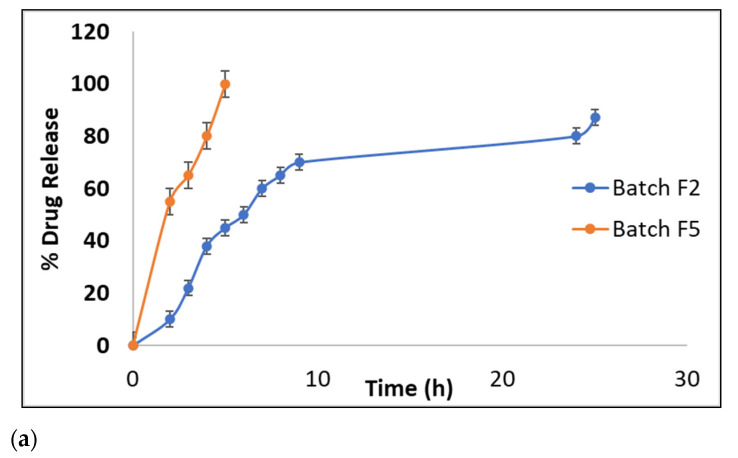
(**a**) The drug release profile from tablets formulated with PM (batch F2) and TRM (batch F5). (**b**) Permeation studies of drug through tablets formulated with PM (batch F2) and TRM (batch F5) compared to plain drug solution.

**Table 1 molecules-27-06829-t001:** The evaluation of buccal tablets of Ivabradine Hydrochloride.

Batches	Hardness (kg/cm^2^)	Swelling Index	Mucoadhesive Strength (gm)	Force of Adhesion (N)
F1	4.5	53.67% ± 1.2 *	54.41 ± 0.5	0.5337
F2	4.5	58.54% ± 1.4 *	57.45 ± 0.6	0.5635
F3	4.8	58.52% ± 1.3 *	60.14 ± 0.4	0.5899
F4	4.5	77.54% ± 1.4 #	79.28 ± 0.3	0.7801
F5	4.8	86.02% ± 1.3 #	99.78 ± 0.2	0.9788
F6	4.8	80.35% ± 1.2 #	90.54 ± 0.4	0.8881

* Swelling index at the end of 6 h; # swelling index at the end of 24 h.

**Table 2 molecules-27-06829-t002:** The formulation of bilayer tablets of ivabradine hydrochloride.

Batch	Polymer	Polymer (mg)	Drug (mg)	Ethyl Cellulose (mg)	Total Weight of Tablet (mg)
F1	PM	59	11	180	250
F2	PM	89	11	150	250
F3	PM	119	11	120	250
F4	TRM	59	11	180	250
F5	TRM	89	11	150	250
F6	TRM	119	11	120	250

## Data Availability

Not applicable.

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
