# Peer review of "Improved Mucoadhesion, Permeation and In Vitro Anticancer Potential of Synthesized Thiolated Acacia and Karaya Gum Combination: A Systematic Study"

_molecules, 2022, doi:10.3390/molecules27206829_

Round 1

Reviewer 1 Report

The manuscript describes the synthesis of thiolated acacia and karaya gums and their anticancer potential. I think it can be published after major revision.

1. Resolution of all figures is very poor. They must be replaced with high resolution figures with resolution at least 400 dpi.

2. The HNMR and CONS of the thiolated polymer using a combination of gum acacia and gum karaya must be provided.

3. Please mention the Synthesis procedure, materials and characterization in a separate section entitled "Materials and Methods' '

4. In Figure 3 the authors mentioned the XRD pattern of gum acacia and gum karaya, but I do not find any XRD regarding both of them.

5. ANOVA analysis must be provided for cytotoxicity and anticancer properties

6. Please dedicate a scheme for showing the chemical reaction between gum and organic compounds.

7. Please change the title of the manuscript.

8. Introduction is poor. Please use of the following references in the introduction:https://doi.org/10.1007/s40097-022-00471-8; https://doi.org/10.1016/j.ijbiomac.2022.01.157;  https://doi.org/10.1016/j.carbpol.2020.116678; 

9. The English level of the manuscript is very poor.

Author Response

The manuscript describes the synthesis of thiolated acacia and karaya gums and their anticancer potential. I think it can be published after major revision.

1. Resolution of all figures is very poor. They must be replaced with high resolution figures with resolution at least 400 dpi.

2. The HNMR and CONS of the thiolated polymer using a combination of gum acacia and gum karaya must be provided.

3. Please mention the Synthesis procedure, materials and characterization in a separate section entitled "Materials and Methods' '

4. In Figure 3 the authors mentioned the XRD pattern of gum acacia and gum karaya, but I do not find any XRD regarding both of them.

5. ANOVA analysis must be provided for cytotoxicity and anticancer properties

6. Please dedicate a scheme for showing the chemical reaction between gum and organic compounds.

7. Please change the title of the manuscript.

8. Introduction is poor. Please use of the following references in the introduction:https://doi.org/10.1007/s40097-022-00471-8; https://doi.org/10.1016/j.ijbiomac.2022.01.157;  https://doi.org/10.1016/j.carbpol.2020.116678; 

9. The English level of the manuscript is very poor.

Answer:

All the changes are made as per reviewers comments. Kindly give permission to upload revised manuscript 

Reviewer 2 Report

The manuscript entitled "Synthesis and characterization of thiolated Acacia and Karaya gum combination for improved mucoadhesion, permeation and in-vitro anticancer potential" is certainly of interest but not publishable in its present form.

To reach the journal standard, I recommend the authors to re-write their manuscript and submit it in an appropriate journal once it will be really ready for reviewing.

I have reported below some comments:

For better clarity, I first invite the authors to completely review their introduction with a special care of the punctuation use. A scheme of the chemical structure of Acacia and/or Karaya gum is really needed. All figures have to be properly replotted using a dedicated software (eg. Origin, QTplot, Kaleidagraph...), a simple screenshot is unacceptable. In FTIR spectra, the axis titles are simply missing, DSC curves and Xray Diffraction spectra are just unreadible. No errors bars in the presented curves of Fig 4 for drug release and drug permeation ... However, it is not just a matter of form but also the chemistry part is not appropriate as acidic conditions (HCl 7N 70°C for 4h) are very corrosive. It necessarily leads to hydrolysis reactions of the natural polymer chains. This hydrolysis should enter in competition with the claimed esterification (no determination of polymer chain lengths to discuss further). The presence of shorter polymer chains may cause the observed enhanced permeation in Fig. 4. Also, regarding the potential use for human health, the methanol use during the synthesis is also very questionable...

In the view of a future submission, I suggest the authors to check the chain lengths of their polymers by Gel Permeation Chromatography or any other appropriate methods before and after esterification. Depending the result, authors may change their title and focus on what they bring a real added-value, the biological tests. If they want to pursue focussing on the chemistry, they should strongly reinforce their chemical reactions description, justify their conditions choice by comparison with other mild and green chemistry routes and strongly increase their characterisation standard.

Author Response

All the changes are made as per reviewers' comments.

Reviewer 3 Report

This work is devoted to the preparation of thio derivatives ACACIA AND KARAYA GUM. The work is written in an understandable and accessible language, although sometimes there are some errors that it is desirable to double-check. Derivatives of food polysaccharides are in the trend of research in Food Chemistry, so this work can be considered modern in terms of the level and scope of research. The advantages of this work are the data of various physical and chemical analyzes, the amount of data, etc. At the same time, there are some important points that must be finalized:

1. Technical points:

1.1. Figure 1 and 2. Poor image quality. This needs to be improved. In addition, the text in these figures is completely unreadable.

1.2. Figure 3. Increase the text of the axis labels. it is unreadable.

1.3. All formulas must be built in the formula editor, but not in plain text. In addition, each formula is usually numbered.

1.4. There are technical errors in the text. It is desirable to read.

2. Main remarks:

2.1. If the authors make a chemical modification of polysaccharides, then it is desirable to give the reaction equation.

2.2. In the introduction, it is also desirable to describe other methods for obtaining sulfur-containing food polysaccharides:

2.3. The data of physicochemical methods of analysis are written too modestly. It is desirable to expand them.

2.4. When describing the experimental data, it is desirable to supplement with references to the literature and compare with data from the literature. This will give more validity to the conclusions of the authors.

2.5. ACACIA GUM is gum arabic?

2.6. What is the role of each component in the mixture? Why was such a system chosen? What functions does Gum acacia perform in the system?

2.7. If you replace Gum acacia with guar, will there be any significant changes?

Author Response

All the changes are made as per reviewers comments

Round 2

Reviewer 1 Report

I am satisfied with changes. Thus it can be published in current form.

Author Response

Thank you very much for reviewing our manuscript.

Reviewer 3 Report

The authors have significantly improved the quality of the article.

Remaining recommendation: please cite article: 10.3390/foods10112571.

Author Response

Point 1. The authors have significantly improved the quality of the article.

Point 2. Please cite article: 10.3390/foods10112571:

Kazachenko, A.S.; Vasilieva, N.Y.; Borovkova, V.S.; Fetisova, O.Y.; Issaoui, N.; Malyar, Y.N.; Elsuf’ev, E.V.; Karacharov, A.A.; Skripnikov, A.M.; Miroshnikova, A.V.; Kazachenko, A.S.; Zimonin, D.V.; Ionin, V.A. Food Xanthan Polysaccharide Sulfation Process with Sulfamic Acid. Foods 202110, 2571. https://doi.org/10.3390/foods10112571

In this article, Sulfated xanthan derivatives have anticoagulant and antithrombotic activity have been synthesized using Sulfamic Acid. Authors did not find any direct co-relation between this article to be cited in the text.